# Characterization of Two Na^+^(K^+^, Li^+^)/H^+^ Antiporters from *Natronorubrum daqingense*

**DOI:** 10.3390/ijms241310786

**Published:** 2023-06-28

**Authors:** Qi Wang, Mengwei Qiao, Jinzhu Song

**Affiliations:** School of Life Science and Technology, Harbin Institute of Technology, Harbin 150080, China; 19b928027@stu.hit.edu.cn (Q.W.); 15244670930@163.com (M.Q.)

**Keywords:** *Natronorubrum daqingense*, NhaC family protein, Na^+^/H^+^ antiporter, extremely halophilic archaea

## Abstract

The Na^+^/H^+^ antiporter NhaC family protein is a kind of Na^+^/H^+^ exchanger from the ion transporter (IT) superfamily, which has mainly been identified in the halophilic bacteria of *Bacillus*. However, little is known about the Na^+^/H^+^ antiporter NhaC family of proteins in the extremely halophilic archaea. In this study, two Na^+^/H^+^ antiporter genes, *nhaC1* and *nhaC2*, were screened from the genome of *Natronorubrum daqingense* based on the gene library and complementation of salt-sensitive *Escherichia coli* KNabc. A clone vector pUC18 containing *nhaC1* or *nhaC2* could make KNabc tolerate 0.6 M/0.7 M NaCl or 30 mM/40 mM LiCl and a pH of up to 8.5/9.5, respectively. Functional analysis shows that the Na^+^(K^+^, Li^+^)/H^+^ antiport activities of NhaC1 and NhaC2 are both pH-dependent in the range of pH 7.0–10.0, and the optimal pH is 9.5. Phylogenetic analysis shows that both NhaC1 and NhaC2 belong to the Na^+^/H^+^ antiporter NhaC family of proteins and are significantly distant from the identified NhaC proteins from *Bacillus*. In summary, we have identified two Na^+^(K^+^, Li^+^)/H^+^ antiporters from *N. daqingense*.

## 1. Introduction

In environments with a high salt concentration, the damage of Na^+^ stress to organisms is mainly manifested as water loss, ion toxicity, and imbalance in osmotic pressure. In order to adapt to the toxicity, microorganisms have formed three major salt tolerance mechanisms: (1) microorganisms change the permeability of their cell membranes to adapt to high-salinity environments by changing the proportion of membrane lipid components in the cell membrane [1,2]; (2) microorganisms accumulate a high concentration of compatible substances within cells as osmotic regulators to maintain osmotic balance [3]; (3) microorganisms extrude Na^+^ from cells while accumulating KCl to maintain an osmotic balance within the cells [4]. These proteins are named Na^+^/H^+^ exchangers (NHEs) [5], can extrude Na^+^, and can be divided into primary Na^+^ pumps [6] and secondary Na^+^/H^+^ pumps [7]. The primary Na^+^ pumps mainly include four types [8]: Na^+^/K^+^-ATPase [9,10], decarboxylase [11,12], NADH-ubiquinone oxidoreductase [13,14], and N5-methyltetrahydromethanopterin: coenzyme M methyltransferase [15,16,17]. The secondary Na^+^/H^+^ pumps are the main Na^+^ efflux system in halophilic bacteria, namely, the Na^+^/H^+^ antiporter, and are the main adaptive response system of halophilic bacteria to Na^+^ stress [18]. At present, the reported Na^+^/H^+^ antiporters can be divided into cation/proton antiporters (CPAs) superfamilies [5], IT superfamilies [19,20], and Na^+^-translocating multiple resistance and pH transporter (Mrp) superfamilies (TC identifier: 2.A.63.1.4) [21]. NHEs have also been widely reported in plants [22] and animals [23].

The NhaC family protein is a Na^+^/H^+^ exchanger belong to the IT superfamily, mainly identified in *Bacillus*. As early as 1997, a Na^+^/H^+^ antiporter encoded by *nhaC* was found, and the gene sequence from Alkaliphilic *Bacillus firmus* OF4 was predicted to encode a membrane protein with 462 residues and 12 transmembrane segments (TMSs). Bf-NhaC was highly homologous to the unknown function products encoded by homologous genes from *Bacillus subtilis* and *Haemophilus influenzae* [24]. The *g1-nhaC* isolated from *Bacillus* sp. G1 could restore the growth of *E. coli* BL21 (DE3) when grown in a medium containing 0.2–2.0 M NaCl, and the Na^+^ efflux activity of g1-NhaC was detected in the pH range 8.0–10.0 [25]. Haja D. K. et al. [26] also reported gene sequences that could encode NhaC proteins from the hyperthermophilic archaeon *Pyrococcus furiosus*.

In our previous study, an extremely halophilic archaeon was isolated from the salinized soil of Daqing City, Heilongjiang Province, China [27]. The strain JX313^T^ was once named *Haloterrigena daqingense* until de la Haba R. R. et al. [28] reclassified it into the genus *Natronorubrum*, and it was renamed *N. daqingense*. *N. daqingense* has extremely strong salt tolerance, and its growth salinity ranges from 10% to saturated (*w*/*v*) (the optimum growth salinity is 17.5%), with a pH range 8.0–11.0 (the optimum pH is 10.0) [27]. Therefore, *N. daqingense* has great potential to screen genes that can encode Na^+^/H^+^ exchangers. In 2022, Wang S. et al. [29] disclosed more detailed data when analyzing the whole genome sequencing of *Natronorubrum daqingense*, which will also be conducive to the identification of Na^+^/H^+^ antiporters. In this study, the salt-sensitive *E. coli* KNabc (*E. coli* with the knockout of three major Na^+^/H^+^ antiporter genes: *nhaA*, *nhaB*, and *chaA*) was used to screen Na^+^/H^+^ antiporter genes from the genome of *N. daqingense*. Two NhaC-type Na^+^/H^+^ antiporter genes were found, which allow *E. coli* Knabc to tolerate 0.6 M/0.7 M NaCl or 30 mM/40 mM LiCl, respectively, with a maximum resistance of pH 8.5/9.5. In summary, we cloned *nhaC1* and *nhaC2* from the *N. daqingense* genome, expressed and functionally validated their encoded proteins, and finally identified two novel NhaC-type Na^+^/H^+^ antiporters from extremely halophilic archaea.

## 2. Results

### 2.1. Cloning and Sequence Analysis of Na^+^(K^+^, Li^+^)/H^+^ Antiporter Genes

To screen for genes that exert an Na^+^/H^+^ antiport function from *N. daqingense* JX313^T^, the genomic library and the salt-sensitive *E. coli* KNabc functional complementary method [30] were combined to select positive clones that could enable KNabc to restore the growth in the medium containing 0.2 M NaCl. Positive clones were grouped based on plasmid electrophoresis and double enzyme digestion results, and two different positive clones [pUC1028-1,9 (pUC18 carrying a 3.5 kb or 4.9 kb fragment)] were obtained, as shown in Figure 1A,B, which enabled *E. coli* KNabc to grow on the LBK medium under the intolerable 0.2 M NaCl concentration. 

Sequence analysis revealed that: (1) a pUC1028-1 DNA fragment (3.5 kb) containing two 5′end-truncated ORFs (ORF1, 2) and two complete ORFs (ORF3, 4), of which only ORF3 (1701 bp) contained the promoter-predicted sequence and Shine-Dalgarno (SD) sequence; (2) a pUC1028-9 DNA fragment (4.9 kb) containing one 5′end-truncated ORF (ORF1) and three complete ORFs (ORF2, 3, 4), with only ORF2 (1575 bp) containing the promoter-predicted sequence and SD sequence. 

In order to screen out ORFs that exert a Na^+^/H^+^ antiport function from positive clones, complete ORFs containing predicted promoter-like sequences were subcloned into pUC18, and complete ORFs without a predicted promoter-like sequence were subcloned into pET28AK (pET28(a) inserted ampicillin resistance gene at *Bgl*II;, as shown in Figure 1C). During the subcloning process, the transformed mixture was coated on the LBK medium (containing 50 mg∙mL^−1^ of ampicillin) plates to obtain positive subclones. Subclones were lined on the LBK medium (containing 0.2 M NaCl and 50 mg∙mL^−1^ of ampicillin) plates to screen for the ORF exerts the Na^+^/H^+^ antiport function, as shown in Figure 1D,E.

As shown in Figure 1D,E, three ORFs could enable KNabc to restore salt tolerance function, that is, two ORFs exhibit Na^+^/H^+^ antiport activity. The gene names of 1028-1-ORF3 and 1028-9-ORF2 will be abbreviated as *nhaC1* and *nhaC2*, and the proteins they may encode will be abbreviated as NhaC1 and NhaC2 in subsequent studies. The characteristics of the hypothetical proteins corresponding to the identified ORFs are listed in Table 1. The protein encoded by 1028-1-ORF3 is predicted to be a membrane protein composed of eleven transmembrane segments (TMSs) with a calculated molecular weight of 60.7 kD. A total of 302 out of 566 amino acids that make up this protein are hydrophobic, resulting in a lower polarity of the entire protein sequence. As a predicted membrane protein, its TMSs include TMS I (4–21), TMS II (28–50), TMS III (65–87), TMS IV (144–166), TMS V (191–213), TMS VI (259–281), TMS VII (320–342), TMS VIII (355–374), TMS IX (399–416), TMS X (486–508), and TMS XI (512–529) (Figure 2C). Another ORF capable of restoring the salt tolerance function of KNabc is 1028-9-ORF2, which is predicted to be a membrane protein with twelve TMSs (Figure 2D), including TMS I (35–57), TMS II (61–83), TMS III (96–115), TMS IV (125–147), TMS X (154–176), TMS VI (186–208), TMS VII (221–243), TMS VIII (263–285), TMS IX (290–312), TMS X (342–364), TMS XI (446–466), and TMS XII (486–508). The translation of the DNA sequence into the amino acid sequence revealed that 301 of the 524 amino acids are hydrophobic, with a calculated molecular weight of 54,785.66 Dalton, resulting in a low polarity of the entire sequence. Based on the characteristics of the Na^+^/H^+^ antiporter, the above two ORFs are likely to possess Na^+^/H^+^ antiport activity [31,32,33,34,35]. The protein encoded by 1028-7-ORF2 could also restore the growth of salt-sensitive *E. coli* KNabc. However, our identification results in subsequent studies showed that it belongs to different types of Na^+^/H^+^ antiporters from the two ORFs in this article.

### 2.2. Phylogenetic Analysis Based on the Neighbour-Joining Algorithm

According to the BlastX comparison results on NCBI [36], both 1028-1-ORF3 and 1028-9-ORF2 were assumed to be Na^+^/H^+^ antiporter NhaC family proteins based on the genome sequencing results; two sequences showed 100% similarity to the published gene sequences BB347_08225 and BB347_08285, respectively, and 1028-1-ORF3 and 1028-9-ORF2 were speculated to be proteins of the Na^+^/H^+^ antiporter NhaC family. To verify this hypothesis and their evolutionary relationships with the identified/predicted proteins with Na^+^/H^+^ antiport activity, a phylogenetic analysis was conducted based on the neighbour-joining algorithm, as shown in Figure 3. Two ORFs were translated into amino acid sequences and then compared using BlastP to select homologs on NCBI [36]. 

1028-1-ORF3 and 1028-9-ORF2 were also aligned with their 11 and 14 sequences producing significant alignments with a percent identity from 80.43% to 96.47% and from 85.11% to 98.28%, respectively, and the similarities and differences between ORFs and their homologs in the composition of amino acid sequences are as shown in Figure 4A,B. 1028-1-ORF3 has the respective identities of from 87% to 100% with closely related homologs from *Natronorubrum sediminis*, *Natronococcus occultus*, and *Natronococcus amylolyticus*, and the Na^+^/H^+^ antiporter NhaC family protein of *Natronorubrum sediminis* shares the bootstap of 100 with 1028-1-ORF3, which means they are sisterly in phylogeny. 1028-9-ORF2 has the respective identities of from 55% to 100% with 21 homologs from *Haloterrigena*, *Halopiger*, *Natrinema*, *Natronorubrum*, *Halostagnicola*, *Euryarchaeota*, *Halakalicoccus*, and *Halovivax*. The Na^+^/H^+^ antiporter NhaC family protein of *Natronorubrum sediminis* shares the bootstrap of 100 with 1028-1-ORF3, which means they are sisterly in phylogeny. 1028-1-ORF3 clustered with some homologs; the 1028-9-ORF2 clustered with all homologs belonging to the Na^+^/H^+^ antiporter NhaC family. 1028-1-ORF3 is relatively close to the identified protein in terms of Na^+^/H^+^ antiport activity, but both 1028-1-ORF3 and 1028-9-ORF2 are significantly distant from it.

According to sequence alignment and phylogenetic analysis, the proteins encoded by 1028-1-ORF3 and 1028-9-ORF2 were Na^+^/H^+^ antiporter NhaC family proteins.

### 2.3. Test for Salt Tolerance and Alkaline pH Resistance

In order to test the salt tolerance of *E. coli* KNabc containing *nhaC1* or *nhaC2*, growth tests were performed in LBK medium (containing 50 mg∙mL^−1^ of ampicillin, pH 7.5) with varying concentrations of NaCl (0–0.8 M) or LiCl (0–50 mM), respectively (KNabc containing pUC18 as a negative control). As shown in Figure 5A,B, KNabc/*nhaC1* could grow up to the presence of 0.6 M NaCl or 30 mM LiCl, and KNabc/*nhaC2* could grow up to the presence of 0.7 M NaCl or 40 mM LiCl, but KNabc/pUC18 could not grow in the presence of 0.2 M NaCl or 5 mM LiCl. KNabc/*nhaC1*, KNabc/*nhaC2*, and KNabc/pUC18 were cultured in an LBK medium (containing 50 mM NaCl, pH 7.0–10.0) to analyze the alkaline pH resistance of *nhaC1* and *nhaC2*. With the gradual increase in pH, the growth of all groups was inhibited to varying degrees, as shown in Figure 5C. Compared with KNabc/pUC18, which was significantly inhibited and almost unable to grow at pH 8.0, both KNabc/*nhaC1* and KNabc/*nhaC2* exhibited strong alkaline pH resistance. Although KNabc/*nhaC1* could still grow at pH 8.5, *nhaC2* could confer higher alkaline pH resistance on KNabc, resisting up to pH 9.5.

### 2.4. SDS-PAGE and Western Blot Analysis of NhaC1 and NhaC2

In order to study the protein localization and function of NhaC1 and NhaC2, two genes were respectively inserted into pET28AK containing a T7 promoter using homologous recombination for protein expression, and the inserted gene sequence would be co-expressed with the 6×His tag. To determine the approximate position of the protein NhaC1 and NhaC2’s SDS-PAGE bands, the everted membrane vesicles from *E. coli* KNabc containing recombinant plasmids were purified by 6×His tag affinity chromatography and used for SDS-PAGE. Combined with the prediction of protein molecular weights in the UniProt database and the peptide molecular weight after translation sites on the pET28AK, the theoretical molecular weights of NhaC1 fusion 6×His protein and NhaC2 fusion 6×His protein should be 64 kD and 58 kD, respectively. As shown in Figure 6A,B, the molecular weights of the fusion proteins after SDS-PAGE were approximately 64 kD and 58 kD, respectively.

Total protein, cytoplasmic protein, and membrane protein were extracted from *E. coli* KNabc containing recombinant plasmids (KNabc containing pET28AK was used as negative control) and used for the Western blot analysis. As shown in Figure 6C,D, both NhaC1 and NhaC2 were only detected from total protein and membrane protein. Combined with previous bioinformatics predictions, it was proven that both of them were located on membranes in cells and were membrane proteins.

### 2.5. Detection of Na^+^(K^+^, Li^+^)/H^+^ Antiport Activity

The Na^+^(K^+^, Li^+^)/H^+^ antiport activities were detected by adding D-lactic acid to the reaction buffer A (pH 7.0–10.0) containing everted membrane vesicles to quench fluorescence (everted membrane vesicles prepared by KNabc/pET28AK-*nhaC1* and KNabc/pET28AK-*nhaC2* as treatment groups, everted membrane vesicles prepared by KNabc/pET28AK as negative control), and then measuring the ratio of dequenching fluorescence after adding various concentrations of NaCl, KCl (Na free), and LiCl. As shown in Figure 7A, Na^+^(K^+^, Li^+^)/H^+^ antiport activities were detected in everted membrane vesicles from KNabc containing pET28AK/*nhaC1* and pET28AK/*nhaC2* when pH 7.5, while no Na^+^(K^+^, Li^+^)/H^+^ antiport activity was detected in everted membrane vesicles from *E. coli* KNabc containing pET28AK. NhaC1 exhibited Na^+^(K^+^, Li^+^)/H^+^ antiport activity when pH ranged from 7.0 to 10.0, with optimal antiport activities at pH 9.5. NhaC2 also exhibited Na^+^(K^+^, Li^+^)/H^+^ antiport activity when pH ranged from 7.0 to 10.0; notably, the Li^+^/H^+^ antiport activity of NhaC2 was lower than that of NhaC1 most of the time, but its Li^+^/H^+^ antiport activity reached its maximum, and was higher than that of NhaC1, at pH 9.5.

### 2.6. Calculation of K_0.5_ Values for Monovalent Cations

To evaluate the substrate affinity activity of NhaC1 and NhaC2 to monovalent cations (Na^+^, K^+^ and Li^+^), the fluorescence-dequenching rates of the everted membrane vesicles were measured at pH 9.5 after adding NaCl, Na-free KCl, or LiCl (final concentration 0.5–10 mM), respectively. According to the data, nonlinear regression analysis was carried out to determine the concentration of monovalent cations that were added when reaching the half-maximum deactivation rate to calculate the value of K_0.5_. After analysis and calculation, it was found that the K_0.5_ values of NhaC1 for Na^+^, K^+^, and Li^+^ were 0.43 mM (Figure 8A), 0.53 mM (Figure 8B), and 0.52 mM (Figure 8C), respectively, indicating that the substrate affinity activity for monovalent cations was Na^+^ > Li^+^ > K^+^. Additionally, the K_0.5_ values of NhaC2 for Na^+^, K^+^, and Li^+^ were 0.42 mM (Figure 8D), 0.39 mM (Figure 8E), and 0.85 mM (Figure 8F), indicating that the substrate affinity activity for monovalent cations was K^+^ > Na^+^ > Li^+^.

## 3. Discussion

In this study, Na^+^(K^+^, Li^+^)/H^+^ antiporter NhaC1 and NhaC2 were identified from *N*. *daqingense*, an extremely halophilic archaeonisolated from Daqing (China) saline-alkali soil, which could grow under a range of NaCl concentrations in 10% saturated solution (*w*/*v*) [27]. Both NhaC1 and NhaC2 belong to the Na^+^/H^+^ antiporter NhaC family protein and are closely related to Na^+^/H^+^ antiporter NhaC family proteins in some halophilic archaea; most of the proteins in this family were identified from halophilic bacteria such as *Bacillus* and verified to have salt tolerance and pH resistance abilities [24,25]. It is interesting that ArcD, an archaebacterial arginine/ornithine antiporter from *Halobacterium salinarum*, was also phylogenetically classified at the edge of Na^+^/H^+^ antiporter NhaC family proteins [37].

NhaC1 and NhaC2 were both predicted to be membrane proteins consisting of 11 and 12 TMSs, respectively (Figure 2C,D), and demonstrated to be localized on the cell membrane of heterogenous host *E. coli* KNabc according to the Western blot results (Figure 6C,D). Both *nhaC1* and *nhaC2* could restore the growth of KNabc in an LBK medium containing 0.2 M NaCl and could be tolerant of higher concentrations of Na^+^ and Li^+^ in the medium, exhibiting stronger pH resistance (Figure 5); these are in line with screening patterns for the Na^+^/H^+^ antiporter complementing of salt-sensitive strain of KNabc [38]. The detection of Na^+^(Li^+^)/H^+^ antiport activities (Figure 7) found that the transportation activities of NhaC1 and NhaC2 would change with changes in pH; it is inferred that their function of transport is pH-dependent. It has been reported that using this method can also identify genes with K^+^/H^+^ antiport function [39] based on the results, both NhaC1 and NhaC2 are inferred to be pH-dependent Na^+^(K^+^, Li^+^)/H^+^ antiporters.

According to available reports [39,40] and TCDB [41], the Na^+^/H^+^ antiporter NhaC (TC identifier: 2.A.35) family protein is a Na^+^/H^+^ exchanger from the IT superfamily, which is characterized by the substrate of the transporter, which should be a charged organic or inorganic chemical species (cations or anions) [19]. Most NhaC family proteins were identified in bacteria [24,25], such as *Bacillus firmus*, *Bacillus subtilis*, *Haemophilus influenziae*, and *Vibrio cholerae*. Notably, genes encoding NhaC proteins have been identified in hyperthermophilic archaea *Pyrococcus furiosus* [26]. Ito et al. [24] identified a NhaC (462 amino acids) containing 12 TMSs from *B. firmus* and detected the Na^+^/H^+^ antiport activity of everted membrane vesicles. The main function of the NhaC family protein is to extrude intracellular Na^+^ or Li^+^ out of cells [24]. For NhaC from *Bacillus* sp. G1, research has shown that NhaC is a Na^+^ extrusion channel at pH 7.5, and ensures pH homeostasis in low Na^+^ environments, reflecting the electrogenic character of the Na^+^/H^+^ antiporter [25]. 

In this study, we identified two proteins with Na^+^(K^+^, Li^+^)/H^+^ antiport activity from *N. daqingense* for the first time; both NhaC1 and NhaC2 belong to the Na^+^/H^+^ antiporter NhaC family of proteins, enriching the research on NhaC-type Na^+^/H^+^ exchangers in extremely halophilic archaea. Therefore, we plan to complete the identification of this protein in future research. We also plan to construct *N. daqingense* mutants with *nhaC1* deletion, *nhaC2* deletion, and *nhaC1*-*nhaC2* co-deletion by using homologous recombination to clarify the role of two NhaC family proteins in the salt tolerance and pH resistance of *N. daqingense* and clarify their mechanisms. 

## 4. Materials and Methods

### 4.1. Strains, Plasmids, and Growth Conditions

The strains and plasmids employed in this study are listed in Table 2. *N. daqingense* JX313^T^ was cultured in 17.5% NaCl (optimum) Luria Bertani (LB) medium containing 10 g/L tryptone, 5 g/L yeast extract, 1.5% glucose (autoclaving alone), 1.23 g/L MgSO_4_∙7H_2_O added to 175 g/L NaCl at 35 °C and pH 9.5. *E. coli* KNabc, the salt-sensitive strain, and its transformant cells were grown aerobically in the LBK medium, that is, the LB medium with 6.49 g/L Na-free KCl instead of NaCl, as previously described by Karpel et al. [38]. The antibiotic concentrations used for screening positive clones were ampicillin 50 μg∙mL^−1^ and kanamycin 50 μg∙mL^−1^. The preparation of electrotransformed competent cells of KNabc required the cultivation of KNabc in LBK medium at 37 °C overnight. In order to test the salt tolerance ability of positive clones, 1% of different KNabc/recombinant plasmid overnight cultures (OD_600_ was adjusted to 0.8 before inoculation) were inoculated into LBK medium (containing 50 mg∙mL^−1^ of ampicillin, pH 7.5) with varying concentrations of NaCl (0–0.8 M) or LiCl (0–50 mM), respectively, and incubated overnight at 37 °C. To determine the pH profile of positive clones, 1% of different KNabc/recombinant plasmid overnight cultures were inoculated into LBK medium (containing 50 mg∙mL^−1^ of ampicillin) with varying pH ranges from 7.0 to 10.0 (adjust by adding 100 mM Hepes–Tris buffer), respectively, and incubated overnight at 37 °C. A UV spectrophotometer was used to measure the OD_600_ value of overnight (16 h) cultures to reflect the growth of recombinant strains. The preparation and electro-conversion method of KNabc electrotransformed competent cells was as described by Jiang et al. [42].

### 4.2. Screening of Na^+^(K^+^, Li^+^)/H^+^ Antiporter Genes

The genomic DNA of *N. daqingense* JX313^T^ was partially digested with the restriction enzyme *Sau*3AI, and the amount of *Sau*3AI was adjusted to concentrate the size of genomic cleavage products between 2–10 kb. Genomic cleavage products prepared after recovery by agarose gel electrophoresis were inserted into the pUC18 (purified after *Bam*HI digestion and dephosphorylation) cloning vector using T4 DNA ligase. The ligation products were transferred into prepared KNabc-competent cells through electro-conversion, and positive clones were selected with LBK medium containing 0.2 M NaCl, 2% Agar, and 50 mg∙mL^−1^ of ampicillin. By using homologous recombination, each ORF and its predicted promoter and SD sequence from positive colonies were respectively ligated to the pUC18 vector; ORFs with no predicted promoter or a low promoter score were ligated into a pET28(a) vector containing T7 promoters. Examples of the above subcloning process are shown in Figure 1C. 

### 4.3. Preparation of Everted Membrane Vesicles

The KNabc containing pUC-positive fragments or pUC18 (as negative control) were inoculated into 100 mL LBK medium (containing 0.2 M NaCl, and 50 mg∙mL^−1^ of ampicillin) and LBK medium (containing 50 mg∙mL^−1^ of ampicillin) at 1% inoculum, respectively, then cultured to the middle of the logarithmic growth stage. These were centrifuged at 5000× *g*, 4 °C for 10 min to collect precipitation, and the supernatant was discarded. After rinsing twice with 10 mL buffer A (140 mM choline chloride; 10 mM Tris; 10% of glycerol; pH 7.5, 1 mM PMSF was added before use; storage at 4 °C), they were suspended and the cell wall was disrupted using a pre-cooled Ultrasonic Cell Disruption System (cycles: 3 s ON followed by 2 s OFF) until the bacterial solution became less turbid. They were centrifuged at 8000× *g*, 4 °C for 10 min, then the supernatant was transferred and ultracentrifuged at 100,000× *g*, 4 °C for 1 h. The supernatant was discarded, and the precipitation was dissolved using the appropriate amount of buffer A, stored at −80 °C.

### 4.4. Preparation and Purification of Proteins

Each ORF that can cause the KNabc to regrow on the LBK medium (containing 0.2 M NaCl) plates was ligated into pET28AK to co-express with the 6×His tag. *E. coli* KNabc containing pET-positive fragments or pET28AK (as negative control) were inoculated into 100 mL LBK medium (containing 0.2 M NaCl, 50 mg∙mL^−1^ of ampicillin) and LBK medium (containing 50 mg∙mL^−1^ of ampicillin) at 1% inoculum, respectively, and cultured at 37 °C. IPTG (final concentration: 1 mM) solution was added when the value of OD_600_ was between 0.6–0.8, and induced the protein expression at 22 °C for 12 h. The total protein, membrane protein, and cytoplasmic protein were prepared by kit from Bestbio (BB-3182&BB-31516). The total protein and cytoplasmic protein were packaged for Western blot analysis, and the rest were stored at −80 °C. The membrane proteins obtained in the previous step were first filtered by 0.22 μm filter membrane, and then affinity chromatography with Ni-NTA using the AKTA protein purifier. The buffer formula involved in the protein purification process is shown in Table 3. The target protein was eluted with 300 mM imidazole, flow velocity: 0.5 mL/min. The target proteins collected by affinity chromatography were transferred into ultrafiltration tubes and centrifuged at 5000× *g*, 4 °C for 20–60 min (until the volume was about 600 μL). The proteins were carefully collected from ultrafiltration and packaged for SDS-PAGE and Western blot analysis.

### 4.5. SDS-PAGE and Western Blot

KNabc/pET28AK-*nhaC1* and KNabc/pET28AK-*nhaC2* were grown in LBK medium containing 0.2 M NaCl and 50 mg∙mL^−1^ of ampicillin; KNabc/pET28AK was grown in LBK medium containing 50 mg∙mL^−1^ of ampicillin as the negative control, and cultured to OD_600_ 0.6–0.8 at 37 °C. Isopropyl-β-D-thiogalactoside (IPTG) was added (final concentration 1 mM) to induce the expression of proteins for 12 h at 22 °C, then used for the preparation and purification of proteins. The extracted protein content was quantified using a Bradford Kit from Bestbio (BB-3411). The methods of SDS-PAGE and Western blot refer to Green et al. [43]. Total protein, cytoplasmic protein, and membrane protein of *E. coli* KNabc (containing pET28AK-positive fragments) and *E. coli* KNabc (containing pET28AK as negative control) were used as samples for detection and localization, respectively. The detection of 6×His tag was carried out using Beyotime polyclonal mouse anti-6×His tag antibody and Beyotime HRP-labeled goat anti-mouse IgG(H+L). 

### 4.6. Detection of Na^+^(K^+^, Li^+^)/H^+^ Antiport Activity

The Na^+^(K^+^, Li^+^)/H^+^ antiport activity was estimated based on the collapse of transmembrane proton gradients, and acridine orange was selected as the fluorescent indicator according to Rosen’s method [44]. A 2 mL buffer D (10 mM Bis-Tris propane; 140 mM choline chloride; 5 mM MgSO_4_; pH from 7.0 to 10.0), 1 μL acridine orange (1 mM), and 40 μg everted membrane vesicles were added into the quartz colorimetric dish, determined after numerical stabilization by fluorescence spectrophotometer with excitation at 490 nm (10 mm slit) and emission at 530 nm (10 mm slit). D-lactic acid (final concentration 5 mM) was added into the mixture to quench the fluorescence; then, various concentrations of NaCl, KCl (Na free), or LiCl were added to dequench the fluorescence, determined after numerical stability. The Na^+^(K^+^, Li^+^)/H^+^ antiport activity was represented by the percentage of fluorescence-dequenching value to total fluorescence quenching value.

### 4.7. Calculation of pH Profile and K_0.5_ Values for Monovalent Cations

The Na^+^(K^+^, Li^+^)/H^+^ antiport activity of different everted membrane vesicles in pH 7.0–10.0 were determined by fluorescence spectrophotometer with NaCl, KCl (Na free) and LiCl at a concentration of 5 mM. Under the optimum pH, equal amounts of NaCl, KCl (Na-free), or LiCl were added with various concentrations (0.5–10 mM) to measure the Na^+^(K^+^, Li^+^)/H^+^ antiport activity of different everted membrane vesicles. With ion concentration as the abscissa and fluorescence dequenching rate as the ordinate, origin2017 was used for nonlinear regression analysis to calculate the K_0.5_ value. 

### 4.8. DNA Manipulation and Bioinformatics Analyses

The extraction of genomic DNA was carried out using a FastPure Bacteria DNA Isolation Kit (Vazyme DC103). The genomic library screening method was described by Green et al. [44]. Preparation of plasmid DNA was carried out using TIANprep Mini Plasmid Kit (Tiangen DP103). Homologous recombination of DNA was done by CloneExpress Ultra One Step Cloning Kit (Vazyme C115-01). The primers used in this study are listed in Table 4. DNA sequencing was performed by RuiBiotech Institute (Beijing, China). The analyses for ORF and constructs of plasmid model diagrams were carried out with SnapGene5.2. Protein and DNA sequence alignment was performed through the National Center for Biotechnology Information using the website https://blast.ncbi.nlm.nih.gov/Blast.cgi (accessed on 1 November 2022), and the accessions of homologs of NhaC1 and NhaC2 to NhaC family proteins used for analysis were listed in Table 5. The phylogenetic tree was constructed via MEGA 11.0 using the neighbour-joining method [45]. Prediction of promoter was performed using the website http://genomes.urv.es/OPTIMIZER/ (accessed on 1 November 2022). The analyses of hydrophobicity and transmembrane prediction conducted using the online analysis tools of DetaiBio’s website http://www.detaibio.com/tools/ (accessed on 15 April 2023).

## Figures and Tables

**Figure 1 ijms-24-10786-f001:**
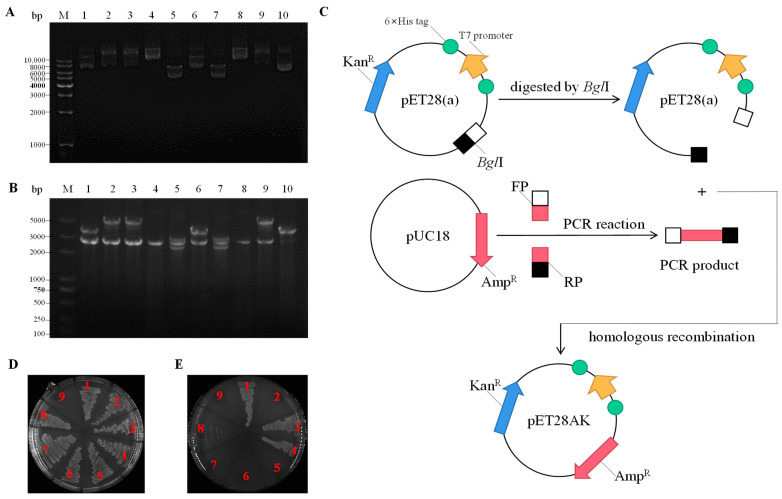
Screening of ORFs that exert Na^+^/H^+^ antiport function. (**A**) The agarose gel electrophoresis of positive clone plasmids that could restore the growth of *E. coli* KNabc on LBK medium containing 0.2 M NaCl; (**B**) The agarose gel electrophoresis after double enzyme digestion of positive clone plasmids by *Eco*RI and *Hin*dIII (the corresponding lanes in A&B are Marker and pUC-1028-1 to pUC1028-10 from left to right); (**C**) Construction schematic of the expression vector pET28AK in this study; (**D**,**E**) *E. coli* KNabc transformants were grown on the LBK medium and LBK medium containing 0.2 M NaCl, (1) KNabc/pUC-1028-1-ORF3, (2) KNabc/pET-1028-1-ORF4, (3) KNabc/pUC-1028-7-ORF2, (4) KNabc/pUC-1028-9-ORF2, (5) KNabc/pET1028-9-ORF3, (6) KNabc/pET1028-9-ORF4, (7) KNabc/pUC18, (8) KNabc/pET28AK, (9) blank.

**Figure 2 ijms-24-10786-f002:**
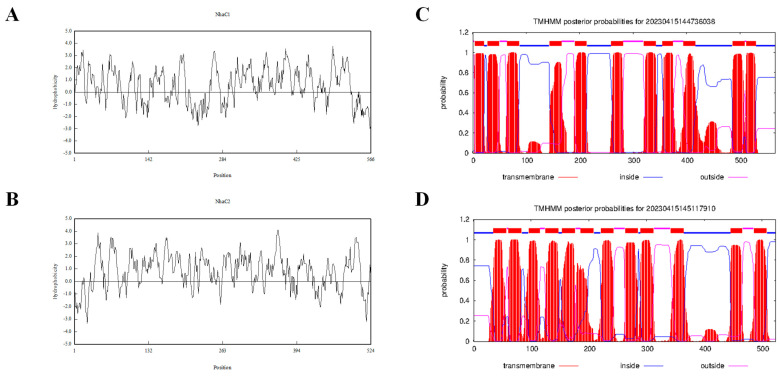
Hydrophobicity analysis and transmembrane segment (TMS) prediction of NhaC1 and NhaC2. (**A**,**B**) Hydrophobicity analysis of NhaC1 and NhaC2; (**C**,**D**) 11 and 12 predicted TMSs of NhaC1 and NhaC2.

**Figure 3 ijms-24-10786-f003:**
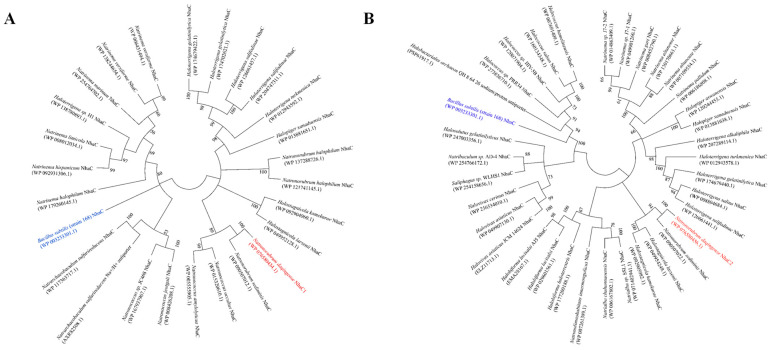
Phylogenetic trees of NhaC1 (**A**) and NhaC2 (**B**) with their closest homologs and an identified protein with Na^+^/H^+^ antiport activity based on the neighbour-joining algorithm. To construct phylogenetic trees, the 24 and 33 closest homologs with 72.84–96.47% and 63.62–98.28% identities, respectively, were selected from 100 sequences. An identical protein from the NhaC family was selected, as shown in blue in Figure 3. Bootstrap values > 50% (based on 1000 replications) are shown at branch points. Both NhaC1 and NhaC2 and their closest homologs clustered with the bootstrap values of 100%; both are shown in bold red in Figure 3.

**Figure 4 ijms-24-10786-f004:**
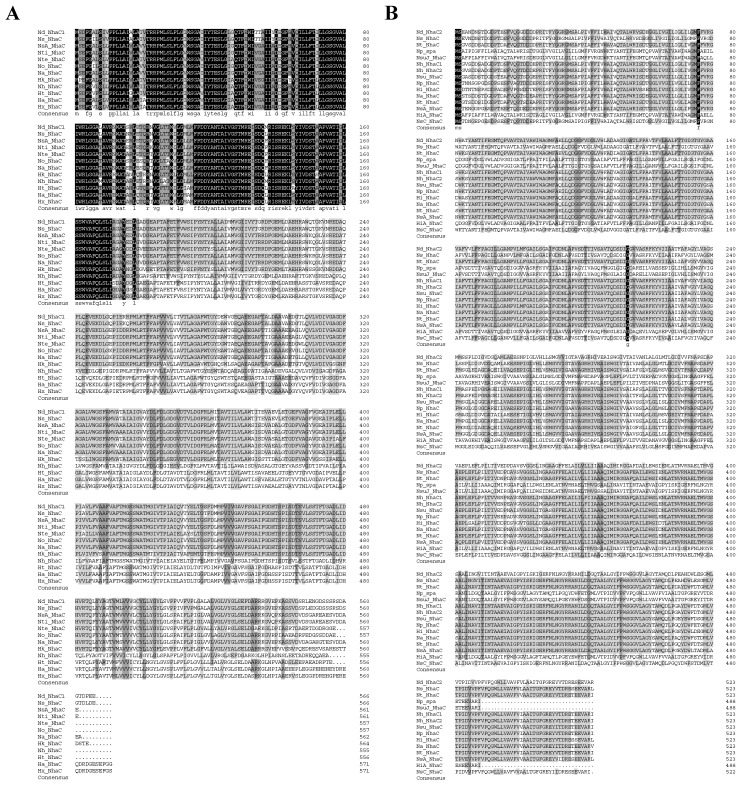
(**A**) Alignment between NhaC1 and homologs of NhaC family proteins. The 11 homologs with percent identities from 80.43% to 96.47% were selected from *Halopiger*, *Haloterrigena, Natronorubrum*, *Halostagnicola*, *Natronococcus*, and *Natronorubrum*. (**B**) Alignment between NhaC2 and 14 homologs of NhaC family proteins. A total of 14 homologs from *Natronococcus*, *Halobiforma*, *Natronorubrum*, and *Halostagnicola* with the percent identity ranging from 85.11% to 93.28% were selected. Shading homology corresponds to 100% (black), ≥75% (grey), ≥50% (light grey), and <50% (white) amino acid identity.

**Figure 5 ijms-24-10786-f005:**
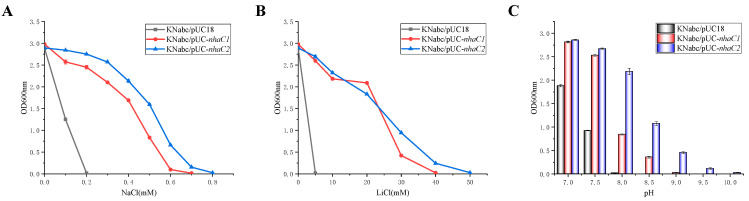
Salt tolerance and alkaline pH resistance of *nhaC1* and *nhaC2*. To test the salt tolerance of KNabc/pUC-*nhaC1* and KNabc/pUC-*nhaC2*, 1% overnight cultures (OD_600_ was adjusted to 0.8 before inoculation) were inoculated to LBK medium containing 0–0.8 M NaCl (**A**) or 0–50 mM LiCl (**B**) and 50 mg∙mL^−1^ of ampicillin, respectively, at pH 7.0, and cultured at 37 °C for 12 h, with KNabc/pUC18 as the negative control, and the OD600 nm was measured; (**C**) 1% overnight cultures (OD_600_ was adjusted to 0.8 before inoculation) were inoculated to LBK medium containing 50 mM NaCl and 50 mg∙mL^−1^ of ampicillin (pH 7.0–10.0) and cultured at 37 °C for 12 h; KNabc/pUC18 was used as negative control, and the OD_600_ was measured. Data in the figure represent the average of three independent trials.

**Figure 6 ijms-24-10786-f006:**
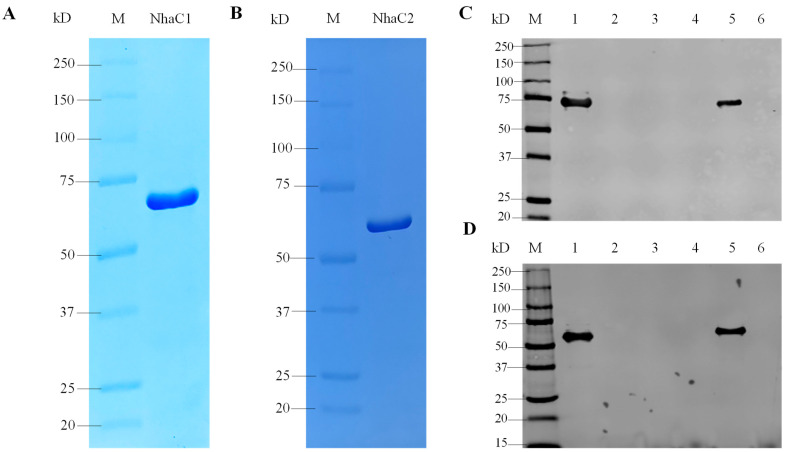
Detection of molecular weights and cell localizations of NhaC1 and NhaC2. (**A**,**B**) SDS-PAGE of NhaC1 fused with 6×His tag and NhaC2 fused with 6×His tag, purified by Ni-NTA affinity chromatography, and the staining method was Coomassie brilliant blue staining; (**C**,**D**) Cell localizations of NhaC1 and NhaC2 detected by Western blots. Total protein, cytoplasmic protein, and membrane protein of NhaC1 fused with 6×His tag and NhaC2 fused with 6×His tag are shown in Lanes 1, 3, and 5, respectively; Total protein, cytoplasmic protein, and membrane protein from KNabc/pET28AK are shown as a negative control in Lanes 2, 4, and 6, respectively.

**Figure 7 ijms-24-10786-f007:**
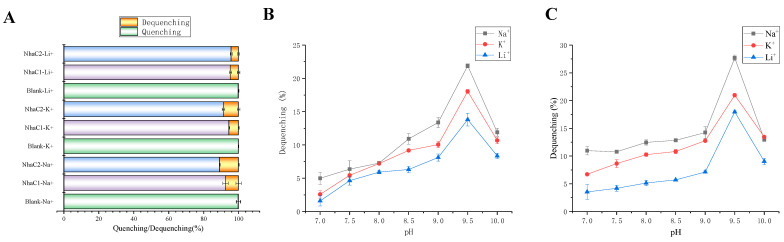
Detection of Na^+^(K^+^, Li^+^)/H^+^ antiport activity. (**A**) The measurements for Na^+^(K^+^, Li^+^)/H^+^ antiport activities were performed in everted membrane vesicles prepared by *E. coli* KNabc containing pET28AK (as negative controls), pET28AK-*nhaC1*, and pET28AK-*nhaC2* at pH 7.0 with monovalent cations (final concentration 5 mM), respectively; (**B**,**C**) Detection of pH-dependent activity profile of NhaC1 and NhaC2. Data in the figure represent the average of three independent trials.

**Figure 8 ijms-24-10786-f008:**
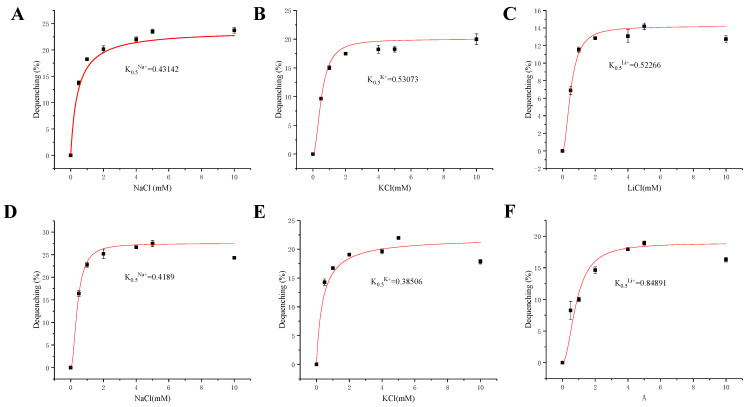
Calculation of K_0.5_ values of Na^+^, K^+^, and Li^+^ by NhaC1 and NhaC2. With the final concentration of monovalent cations as the abscissa and the fluorescence dequenching rate as the ordinate, origin2017 was used for nonlinear regression analysis to fit the curve and calculate the final concentration on monovalent cations added, which could reach half of the maximum fluorescence dequenching. (**A**–**C**) The curves and K_0.5_ values of Na^+^, K^+^, and Li^+^ NhaC1, respectively; (**D**–**F**) The curves and K_0.5_ values of Na^+^, K^+^, and Li^+^ NhaC2 respectively. Data in the figure represents the average of three independent trials.

**Table 1 ijms-24-10786-t001:** Characteristics of the hypothetical proteins corresponding to the identified ORFs.

Characteristics	1028-1-ORF3	1028-9-ORF2
Gene ID	BB347_RS08225	BB347_RS08285
Gene abbreviated name	*nhaC1*	*nhaC2*
Protein abbreviated name	NhaC1	NhaC2
Accession	WP_076580434.1	WP_076580456.1
Amino acid sequence length(aa)	566	524
Number of hydrophobic amino acids (aa)	302	301
Predicted molecular weight (Da)	60,722	54,786
Number of TMSs	11	12
Definition	Na^+^/H^+^ antiporter NhaC family protein	Na^+^/H^+^ antiporter NhaC family protein

**Table 2 ijms-24-10786-t002:** Strains and plasmids employed in this study.

Strain or Plasmid	Description	Source or Reference
Strains		
*N.daqingense* JX313^T^	Original strain, an extremely halophilic archaea	Isolated and identified by our lab [27]
*E. coli* DH5α	Host strain for cloning	Vazyme Biotech Co., Ltd.
*E. coli* KNabc	Na^+^/H^+^ antiporter-deficient strain, *nha*A::Km^R^, *nha*B::Em^R^, *cha*A::Cm^R^	Donated by Prof. Juquan Jiang [31]
Plasmids		
pUC18	Cloning vector, Amp^R^	Comate Biosciences Co., Ltd. (Changchun, China)
pET-28a	Prokaryotic expression vector, Kan^R^	Comate Biosciences Co., Ltd.
pET-28AK	Prokaryotic expression vector, Kan^R^ and Amp^R^	This study
pUC1028-1	pUC18 carrying 3.5 kb DNA fragment with Na^+^/H^+^ antiport activity	This study
pUC1028-9	pUC18 carrying 4.9 kb DNA fragment with Na^+^/H^+^ antiport activity	This study
pET28AK-*nhaC1*	Heterologous expression vector of *nhaC1*	This study
pET28AK-*nhaC2*	Heterologous expression vector of *nhaC2*	This study

**Table 3 ijms-24-10786-t003:** Protein purification buffer formula.

Component	Soluble Film Buffer	Binding Buffer	Washing Buffer	Elution Buffer
Choline chloride	140 mM	140 mM	140 mM	140 mM
Tris	25 mM	25 mM	25 mM	25 mM
Glycerol	10%	10%	10%	10%
N-Dodecyl-β-D-maltoside	0.02%	0.02%	0.02%	0.02%
Imidazole	-	10 M	25/45/50/55/60/65/85 mM	300 mM

**Table 4 ijms-24-10786-t004:** Primers of this study.

Primers	Description	Sequence(from 5′ to 3′)	Source or Reference
22F	Archaea 16S rDNA	ATTCCGGTTGATCCTGC	X.W.Xu, et al. [46]
1540R		AGGAGGTGATCCAGCCGCAG	
M13-47F	Sequencing primers of pUC18	CGCCAGGGTTTTCCCAGTCACGAC	This study
M13R		CACACAGGAAACAGCTATGAC	This study
T7	Sequencing primers of pET-28a and pET28AK	TAATACGACTCACTATAGGG	This study
T7t		GCTAGTTATTGCTCAGCGG	This study
Amp-F	To insertAmp^R^ into pET-28a	CTGCHCGTTGGTGCGGATATCCGCGGAACCCCTATTTGTT	This study
Amp-R		GTATCCCACTACCGAGATATCTTACCAATGCTTAATCAGTGAGGC	This study
1-3FP	To insert DNA sequence of pUC1028-1-ORF3 into pUC18	TATGACCATGATTACGAATTCATGTCTGACTTTGGAGCGCTTT	This study
1-3RP		CAGGTCGACTCTAGAGGATCCTTACTCCTCAGGGTCCGTCCC	This study
9-2FP	To insert DNA sequence of pUC1028-9-ORF2 into pUC18	TATGACCATGATTACGAATTCATGAGTGAAGCCAACGATAATTCA	This study
9-2RP		CAGGTCGACTCTAGAGGATCCTCATAGTCGTGCCACCTCCTCG	This study
NhaC1-EF	To insert *nhaC1* into pET28AK	CAGCAAATGGGTCGCGGATCCATGTCTGACTTTGGAGCGCTTT	This study
NhaC1-ER		TTGTCGACGGAGCTCGAATTCTTACTCCTCAGGGTCCGTCCC	This study
NhaC1-EF	To insert *nhaC2* into pET28AK	CAGCAAATGGGTCGCGGATCCATGAGTGAAGCCAACGATAATTCA	This study
NhaC1-ER		TTGTCGACGGAGCTCGAATTCTCACACCCCCCAGAAGAACG	This study

**Table 5 ijms-24-10786-t005:** Accessions of homologs of NhaC1 and NhaC2 to NhaC family proteins used for analysis.

NhaC1 Homologs	Accession	NhaC2 Homologs	Accession
Nd_NhaC1(this study)	WP_076580434.1	Nd_NhaC2(this study)	WP_076580456.1
Ns_NhaC	WP_090507012.1	Ns_NhaC	WP_090507022.1
NsA_NhaC	WP_278304797.1	Nt_NhaC	WP_076607379.1
Nti_NhaC	WP_006090653.1	Np_spa	TYL39413.1
Nte_NhaC	WP_090303449.1	NsuJ_NhaC	ELY48302.1
No_NhaC	WP_015320610.1	Nh_NhaC1	WP_170972344.1
Na_NhaC	WP_005555905.1	Nh_NhaC2	WP_162989723.1
Hk_NhaC	WP_092904960.1	Nsu_NhaC	WP_049890100.1
Nh_NhaC	WP_137288726.1	Np_NhaC	WP_187432893.1
Ht_NhaC	WP_012943592.1	Hi_NhaC	WP_049954249.1
Ha_NhaC	WP_120244443.1	Na_NhaC	WP_152938558.1
Hx_NhaC	WP_013881651.1	Nt_NhaC	WP_006090630.1
		NsA_NhaC	WP_278304817.1
		HiA_NhaC	EMA28167.1
		NsC_NhaC	WP_252487510.1

## Data Availability

All data are contained within this manuscript.

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
