# Peer review of "Characterization of Two Na^+^(K^+^, Li^+^)/H^+^ Antiporters from *Natronorubrum daqingense"

_ijms, 2023, doi:10.3390/ijms241310786_

Round 1
Reviewer 1 Report
The authors identified and partially characterized two
cation/H+ antiporters of the NhaC family from the archaeon
Natronorubrum daquingense. This reviewer explicitely appreciates
that these proteins have been functionally characterized and
supports that these results should be published. Unfortunately,
there are major quality issues with respect to the writing of
this manuscript, which preclude acceptance in the current form.
(A)
(A.1) In the title, the species name is incorrectly given as
Natronorubrum daquingensis (instead of daquingense), probably because
the species had initially been named Haloterrigena daquingensis.
The species has been reclassified to Natronorubrum daquingense
(PMID:34690986).
(A.2) line 58. Also, the authors invalidly write that the species was
reclassified as "Natronorubrum daquingensis" while in fact it was
reclassified as "Natronorubrum daquingense". This is the same error
as in the manuscript title.
(B)
(B.1) The complete genome sequence of the "Haloterrigena daqingensis"
was reported (PMID:34245190). Unfortunately, those authors ignored the
fact that the species has been reclassified.
(B.2) Why is the genome sequencing paper not cited in the current
manuscript and why are the newly identified (and sequenced) genes not
correlated to the genes in the complete genome? There is a one_to_one
correlation (1028-1-ORF3 is BB347_08225, 1028-9-ORF2 is BB347_08285).
(B.3) The Data Availability Statement "All data are contained within
this manuscript" would not apply, even though the sequences are shown
in Fig.4 A and B. It is mandatory that newly sequenced genes are
submitted to the databases (e.g. GenBank). This would, however, not
be required if it is explicitely stated that they are identical to those
from the published complete genome sequence.
(C)
The manuscript contains several content errors which seem to
originate from typographical errors but lead to statements which
are simply incorrect:
(C.1) line 10 "identified in the halophilic archaea of Bacillus".
Bacillus is a bacterium, not an archaeon.
(C.2) line 32-33: "extrude Na+ from cells while accumulating NaCl
to maintain..." When you extrude Na+ and accumulate Na+, nothing
changes. Cells accumulate KCl, not NaCl.
(D)
Several of the cited references are simply invalid or may not be best
suited for the statement they are cited for. With that number of
erroneous reference citations, the Introduction is not suitable
for publication.
(D.1) line 28 "changing the permeability of their cell membrane ...
by changing the proportion of the membrane lipid components". The cited
ref_1 does not address membrane lipid components, but deals with EPS
(exopolysaccharides).
(D.2) for NHE, the authors cite Gardner et al which, however,
specifically analyzes a single gene product that belongs to NHE11.
PMID:15643048 ("Evolutionary origins of eukaryotic sodium/proton
exchangers") seems a better choice.
(D.3) for the statement "can extrude Na+ and can be divided into primary
Na+ pumps and secondary Na+/H+ pumps", ref_7 is cited. It seems that
an incorrect citation is inserted into the literature list as ref_7.
Curiously, the same publication occurs twice in the reference list,
once as ref_7, once as ref_40. The provided ref_7 deals with Mrp and
I could not find information on the distinction between primary
Na+ pumps and secondary Na+/H+ pumps in that paper.
(D.4) ref_7 is again cited later (line 52), but also invalidly. It is
stated to be Dominik et al, while the authors of currently provided
ref_7 are Haja et al. The literature list currently does not have
any publication from Dominik et al. The cited topic is
Pyrococcus furiosus NhaC, which is not a subject in the
Haja et al paper.
(D.5) ref_7 is also cited for the statement "Most NhaC family proteins
were identified in bacteria" in line 314, which is invalid given the
publication currently provided as ref_7. In line 316 the same
publication is cited as ref_40 for another topic.
(D.6) It is stated that primary Na+ pumps includes four types, the
first being referred to as "ATPase". Ref_9 is a review on
Na+/H+ Exchangers with no detectable reference an any ATPase. Also
ref_10 and ref_11 seem completely unrelated to any ATPase. It thus
remains enigmatic what the authors refer to with the term "ATPase".
Do they want to refer to the ATP-dependent Kdp transporter
(ref_12, associated with "decarboxylase").
(D.7) the 4th type is stated to be methyltetradromethanopterin. This must
be a typo of methyltetra-HYdro-methanopterin. This is linked to ref_17,
which, however, has just nothing to do with that subject.
(D.8) for the statement that Na+/H+ pumps are the main Na+ efflux system
in halophilic bacteria, two references are cited. Ref_18 is OK.
However, why is a description of the TCDB database cited to that statement.
The specific topic (Na+/H+ pumps) seems not to be addressed in the paper
describing the TCDB database, but of course might be addressed in the
database itself. However, then the TCDB category should be provided.
(D.9) Similarly, the publication describing the 2016 update to TCDB is
referenced for Mrp (ref_23), while the only mentioning of Mrp in that
paper is the statement that 3 families are contained within TCDB.
Here, again, the TCDB number for Mrp should be provided.
(D.10) Ref_24 is cited for the statement that "NHEs of the IT superfamily
have also been widely reported in plants". The authors of that paper
address the CPA superfamily, not the IT superfamily.
(D.11) line 76: ref_28 is cited for E.coli KNabc and this citation is
most likely incorrect. The same reference is cited in line 57 for
reclassification of the species, and that citation is in a sense
correct. However, it is referred to as "Oren et al." but the paper
is "de la Haba et al".
(E) Key to the manuscript is the antiport activity, which is
done "according to Rosen's method" (ref_43).
(E.1) Because ref_43 addresses "ion extrusion systems" and has
been published in Methods in Enzymology, the method is assumed by
this reviewer to be suitable for the analyses described. However, I did
not further validate this by accessing and reading that methods paper.
(E.2) line 417: the authors state to have added "40 ug membrane protein".
They actually must have added "an amount of everted membrane vesicles
which corresponds to 40 ug membrane protein". Because the preceding
method (4.4) describes purification of proteins, it should be made
clear that everted membrane vesicles were used, not purified proteins.
(E.3) the term "everted membrane vesicles" is also used for B.firmus OF4
(line 318, ref_25). In that paper, a distinct method is described, not
"Rosen's method".
(F) addition of the His-tag
(F.1) lines 212-214: the authors should make it clear that ORF cloning
into their pET28AK expression vector adds a His-tag to the cloned ORF.
(F.2) lines 389-390: this should also be mentioned in the methods.
(G) lines 215 ff: "the everted membrane vesicles ... were purified
by 6xHis tag affinity chromatography".
(G.1) Is there any information about the average diameter of the
everted membrane vesicles? They must be tiny. Otherwise, a higher
contamination with other membrane proteins, colocalized in the
same membrane patch and thus vesicle, is expected for this comparably
simple purification procedure.
(G.2) Fig.6 A and B: this seems to be Coomassie staining. That should
be mentioned in the legend.
(G.3) Fig.6 C and D: probably, panel C is for nhaC1, panel D for nhaC2.
But that is not stated in the legend.
(H) Fig.1 panels D and E
(H.1) lines 105-107: in the legend to Fig.1 panels D and E, the authors
assign numbers to the analyzed samples. These numbers are not shown
in the figure itself. And there is no mentioning how the numbers#
relate to the figure. Readers will be able to figure this out
themselves (9 is blank), but it is preferrable that the correlation
is made clear in the legend.
(H.2) line 109ff: it is stated that "two ORFs could enable". However,
the figure shows that "three ORFs could enable". Only two of them
(#1, #4) are further characterized in the current manuscript. That
should be clearly stated.
(H.3) for the third ORF (#3, 1028-7-ORF2), no data are reported in
the current manuscript. However, in that case it is highly atypical
to provide results (size, membrane localization, antiport activity)
in the discussion (lines 326-330).
(I) line 112: ORF 1028-1-ORF3 (NhaC1) is stated to be 6.07 kD. That
would be a VERY SHORT protein. NhaC1 may have 60.7 kD as the
His-tagged version is stated to have 64 kD (line 220).
(J) line 124: "Based on the characteristics of Na+/H+ antiporter, the
above two ORFs are likely to possess Na+/H+ antiport activity". Is that
concluded just from the fact that the proteins have 11 or 12 TM domains?
Not all proteins with that number of TM domains are antiporters.
Such a statement is only suitable upon sequence comparison to
NhaC sequences.
(K) B.subtilis malate-2H(+)/Na(+)-lactate antiporter
(K.1) line 48: the authors present many details taken from ref_25,
in which the Bacillus firmus OF4 NhaC was characterized. One cited aspect
reads "... was highly homologous to the unknown function products encoded
by homologous genes from Bacillus subtilis and ...". In ref_25, the
Uniprot accession of the B.subtilis protein is provided (P54571,
misclamed to be a GenBank accession no). That protein has since been
characterized ("malate-2H(+)/Na(+)-lactate antiporter", based on
PMID:10903309, published 2000, where the name is
"malic/Na+-lactate antiporter"). It seems misleading to still refer to
that B.subtilis protein as being uncharacterized, even though that was
the case when ref_25 was published (1997).
(K.2) legend to Fig3: "with their closest homologs and two identical
proteins with Na+/H+ antiport activity". These two proteins
(UniProt accessions O07553 and P54571) are NOT identical (they have
32% protein sequence identity), and only one of them is a
Na+/H+ antiporter. The other is a malate-2H+/Na+-lactate antiporter
(see also stated the annotation provided in the figure itself).
(K.3) line 321 "reflecting the electrogenic character of Na+/H+ antiporter
[ref_38]". Ref_38 describes a malic/Na+-lactate antiporter according to
the title of the cited reference (malate-2H+/Na+-lactate antiporter
according to UniProt), not a Na+/H+ antiporter. The sentence starts,
however, with "For NhaC1 from Bacillus sp. G1", and for that, ref_26
is cited in line 52.
(L) Both, NhaC1 and NhaC2 show 100% sequence identity to a protein
from Natronorubrum sediminis. Is it typical that proteins from
N.daquingense and N.sediminis are (nearly) identical? Do these
species share a common segment or are they thus closely related
throughout their genome?
(M) lines 189-191:
(M.1) "could grow in" might better be phrased "could grow up to".
The mentioned concentrations are the highest which still allowed
some residual growth.
(M.2) The text and figure are inconsistent to each other. According
to the text, nhaC1 has a higher salt tolerance (0.7M Na, 40 mM Li)
than nhaC2 (0.6M Na, 30 mM Li). According to the figure, it is
the other way round.
(N) pH dependence of growth. The data are shown in Fig.5C and
described in lines 197-199. The methods are described in
lines 352-354. Here, it is relevant that the authors confirm that
the alkaline pH was stable over the cultivation period by
measuring pH at the end of the incubation period.
(O) Fig. 7 B and C
(O.1) why is the zero value on the X-axis for C but above the X-axis for B?
(O.2) as I interpret the figure, identical results were obtained for
nhaC1 and nhaC2. Why is the text (lines 253-256) different for the two
paralogs? What does "but the Li+/H+ antiport activity" mean in line 255?
(P) line 275: given the scatter in the raw data, I would question
that the differences between Na+/K+/Li+ are significant for nhaC1.
The affinity seems nearly the same for the three ions. Even for
nhaC2 (line 277), probably only the affinity to Li+ is clearly
reduced.
(Q) line 385: "ultracentrigued at 39000 rpm". This information has
no meaning unless the rotor type is given. Alternatively, provide
the g value.
(R) line 403 "was absorbed from ultrafiltration carefully".
Adsorbe means to attach the protein to the ultrafiltration
tubing, but the authors probably mean the opposite, namely
to liberate the protein.
This manuscript needs a major improvement with respect to the English language.
Author Response
Dear Editor Sureerat Namken, Dear Reviewer1:
Thank you for your comments on June 8. We were pleased to know that our work was rated as potentially acceptable for publication in IJMS, subject to adequate revision. We thank the editor and reviewers for the time and effort that they have put into reviewing the previous version of the manuscript. Your suggestions have enabled us to improve our work. Base on the instructions, we uploaded the file of the revised manuscript. Accordingly, we have uploaded a copy of the original manuscript with all the changes highlighted by using the track changes mode in MS Word.
Appended to this letter is our point-by-point response to the comments raised by the Reviewer1. The comments are reproduced, and our responses are given directly afterward in a different color(red).
Regarding the language, to make it easier for readers to read this paper and remove grammatical and word errors without changing the original meaning, we selected the MDPI editing service to edit our manuscript, and the specific changes can be found in the track changes mode in MS Word.
We would like also to thank you for allowing us to resubmit a revised copy of the manuscript. We hope that the revised manuscript is accepted for publication in the International Journal of Molecular Sciences.
Sincerely,
Jinzhu Song

Reviewer 2 Report
In this manuscript, the authors describe the identification for the first time of two antiporters from Natronorubrum Daqingensis.
Although the work is interesting, several improvements are needed.
Abstract:
· The abstract is very confusing, especially in the introductory part, I suggest rephrasing the sentences to make it more fluent and impactful
Introduction:
1. line 27, imbalance of what?
2. line 33-37 it is necessary to make this classification clearer, perhaps by using a graphical aid such as a table, and in addition, since the second part refers to the Na+/H+ systems typical for halophiles, it would be appropriate to specify in which organisms they are found
3. line 43 is not relevant and can be deleted
4. line 44 Rewrite the sentence by changing it to 'The NhaC family protein is a Na+/H+ antiporter belonging to the IT family'.
Results:
1. Line 71-82 I suggest moving all experimental details to the materials and methods section
2. Line 112 is 6.07kDa, correct?
3. Line 110-114 please make this section clearer
4. 2.1. Cloning and sequence analysis of Na+(K+,Li+)/H+ antiporter genes: Again, I suggest adding a table summarising the ORFs identified, the characteristics of the hypothetical proteins identified, such as the number of amino acids, number of transmembrane domains, hydrophobic amino acid content, etc.
5. Figure 4. Section a and b of the figure can be removed as it gives nothing to the manuscript; if the authors wish to keep this part, I suggest moving it to the supplementary materials. It would be interesting to analyze the percentage of identity between the two proteins NhaC1 and NhaC2.
6. provide the access number of the sequences used in this paragraph (2.2) in the materials and methods section
7. Line 217-221 this section needs to be rewritten more clearly
8. caption figure 6: there are too many unnecessary experimental details, move them to the materials and methods section
9. in any case, the SDS-page alone is not sufficient for the precise determination of molecular weight, I suggest lowering enthusiasm for the precise determination of weight, unless additional experiments, such as a gel filtration in the presence of a molecular weight calibration curve are provided
Discussion:
1. in the revision process of the paper, I suggest that the discussion section be merged with the results section to make it clearer and easier to read; furthermore, the section from line 323 should be separated into a conclusion paragraph which it would also be interesting to add future perspectives
Materials and Methods:
1. Table 1: when possible it would be required to add the connected reference

Some sections of this manuscript are difficult to read, I suggest a thorough revision of the English to make it more fluent
Author Response
Dear Editor Sureerat Namken, Dear Reviewer2:
Thanks very much for taking your time to review this manuscript. I really appreciate all your comments and suggestions! Please find my itemized responses in the appendix and my revisions/corrections in the re-submitted files.
Appended to this letter is our point-by-point response to the comments raised by the Reviewer1. The comments are reproduced, and our responses are given directly afterward in a different color(red).
Regarding the language, to make it easier for readers to read this paper and remove grammatical and word errors without changing the original meaning, we selected the MDPI editing service to edit our manuscript, and the specific changes can be found in the track changes mode in MS Word.
We would like also to thank you for allowing us to resubmit a revised copy of the manuscript. We hope that the revised manuscript is accepted for publication in the International Journal of Molecular Sciences.
Sincerely,
Jinzhu Song

Reviewer 3 Report
The manuscript of Qi Wang and co-authors (Characterization of two Na+(K+, Li+)/H+ antiporters from Natronorubrum Daqingensis) deals with the identification of two Na+/H+ antiporter systems from the halophilic archeon N. drawingensis. Both genes were cloned and transformed in E. coli. Here, screening within a salt sensitive E. coli strain with mutations in three antiporter systems were performed. The localization as well as a first characterization of the corresponding proteins were performed in E. coli as well.
Beside this straight forward approach to characterize both antiporter systems from a halophilic archaeon, there are some points to be consider: Title: Natronorubrum Daqingensis should be written as Natronorubrum daqingensis Lane 10: Bacillus strains do not belong to the kingdom of archaea, they are bacteria Lane 20: delete the last sentence of the abstract. It does not make any sense.Lane 33: “while accumulating NaCl to maintain osmotic balance within the cells”: definitely not: they accumulate KCl.
Lane 76: “The mixture was transferred into E.coli KNabc (nhaA::KmR, nhaB::EmR, 74 chaA::CmR, lacking three major Na+/H+ antiport genes caused cannot grow in the environ-75 ment containing 0.2M or more NaCl)[28] by 42°C thermal shock”: the citation is wrong.
Lanes 83 to 125: the description of the cloning strategy is confusing and should be re-written. Fig. 1 should be re-organized: part C can be deleted, only parts are showing, primers are not indicated as in the primer list and in total, and these kinds of illustrations are old-fashioned and not necessary. Figures 1A and B can be deleted, due to the bad quality, there are no significant findings from these illustrations.
Figure 4: due to the quality of the presentation, the data of this figure should be given as a table with informations about the similarities of the antiporters to other proteins. In addition the accession numbers of the genes reported here should be indicated in the text.
Figure 5: why are there two different ways of displaying of the results. As indicated in the methods part, 1 % of the overnight cultures were inoculated. Did they reach the same optical densities or were the densities adjusted? However, growth curves of the different strains would give more informations about the expression of both genes in E. coli.
In general the characterization of both proteins depends on its expression of the corresponding genes in E. coli and no experiments in N. drawingensis were performed. Therefore, at least the authors should show the production of the proteins in N. drawingensis. A simple westernblot with antibodies against the proteins would be sufficient.
Through the text: please control the larger and smaller letters.
Author Response
Dear Editor Sureerat Namken, Dear Reviewer2:
We highly appreciate all your professional review work on our article and your comments are constructive to show our work in this article. According to your suggestions, we have completed all revising in our previous manuscript, the detailed corrections are listed in the attachment. The comments are reproduced, and our responses are given directly afterward in a different color(red).
Regarding the language, to make it easier for readers to read this paper and remove grammatical and word errors without changing the original meaning, we selected the MDPI editing service to edit our manuscript, and the specific changes can be found in the track changes mode in MS Word.
It is worth mentioning that we have already revised the manuscript according to the comments of other reviewers when we received your comments, and there are only two days left before the deadline for reply. Although the time is short, we still try our best to make changes, and sincerely reply to each of your comments.
Thank you again for allowing us to resubmit a revised copy of the manuscript. We hope that the revised manuscript will be accepted for publication in the International Journal of Molecular Sciences.
Sincerely,
Jinzhu Song

Round 2
Reviewer 1 Report
-
Editing of the English language is required.
Author Response
Dear Editor Sureerat Namken and Reviewer1:
Thank you for giving us the opportunity to revise the manuscript after the second round of review. We have completed the revision of the content in the manuscript according to the reviewer's requirements, and explained the reviewer's questions.
We would like to express our sincere gratitude to you for finding such a conscientious reviewer, who gave detailed and accurate comments, with a high degree of professionalism, and we would like to express our admiration for the patience and care shown by the reviewers during the review process. We have made changes and responded in the hope that our response will be satisfactory to reviewers and editors.
All three reviewers praised the scholarly content of the manuscript.To make up for our shortcomings in English writing, we sent the manuscript to MDPI language editing service for polishing, so that the accuracy of the wording and grammar of this manuscript has been greatly improved, and it has met the standards for publication in academic journals.
We express our heartfelt thanks and respect to all those who worked on this manuscript. We hope that our revised manuscript can be published in IJMS. Please contact us if you have any questions. Thank you!
Best wishes,
Jinzhu Song

Reviewer 2 Report
I thank the authors for responding to my comments. I have nothing further to add
Author Response
Thank you for your recognition of the quality of our manuscripts and academic achievements, each of your comments is valuable, each of your suggestions is precious to us, and the quality of the manuscripts revised according to your suggestions has been improved. Thank you again for your effort in reviewing this manuscript.
Reviewer 3 Report
The manuscript of Qi Wang and co-authors (Characterization of two Na+(K+, Li+)/H+ antiporters from Natronorubrum daqingensis) can be published in its present form.
Author Response

(The authors gave the same response as above.)
